# Noise Evaluation of Coated Polymer Gears

**DOI:** 10.3390/polym15030783

**Published:** 2023-02-03

**Authors:** Brigita Polanec, Srečko Glodež, Aleš Belšak

**Affiliations:** Faculty of Mechanical Engineering, University of Maribor, Smetanova 17, 2000 Maribor, Slovenia

**Keywords:** polymer gears, PVD-coatings, experimental testing, noise evaluation

## Abstract

A comprehensive experimental investigation of the noise evaluation of coated spur polymer gears made of POM was performed in this study. The three Physical Vapour Deposition (PVD) coatings investigated were aluminium (Al), chromium (Cr), and chromium nitrite (CrN). The gears were tested on an in-house-developed testing machine under a torque of 20 Nm and at a rotational speed of 1000 rpm. The noise measurements were performed with the tested gear pair on the testing device with a sound-proof acoustic foam used for the acoustic sound-proof insulation. The sound signal was analysed in time, frequency, and time–frequency domains and typical phenomena were identified in the signal. Experimental results showed that the noise level was higher for polymer gears with different coatings if compared to the polymer gears without coatings. With sound analysis in the time–frequency domain, precise degradation of the coatings could be noticed. In future studies, it would be appropriate to use a new method for signal analysis, e.g., high-order statistics and hybrid technique.

## 1. Introduction

Polymer gears are used widely in many engineering applications, such as office appliances, mechatronic devices, household facilities, and medical instruments [1,2,3]. They can operate without lubrication and, therefore, may be used in some typical applications where lubrication is not desired (household appliances, the food industry, medicine, etc.). As presented by Zorko et al. [4], polymer gears dampen vibrations better and exhibit a better response to noise and vibration if compared to metal gears. Furthermore, polymer gears are mostly resistant to corrosion and other chemical influences, and, consequently, can operate in environments where corrosive substances are present. In general, polymer gears can be produced by classical cutting processes or, for large series production, by injection moulding [5,6]. Other benefits of polymer gears are also the high size–weight ratio, low coefficient of friction, high resistance against impact loading, etc. [7,8,9]. As presented by Hribersek et al. [10], polymer gears started to be used in a variety of power transmissions applications, including demanding high-performance uses in products with high added values. On the other hand, polymer gears also have some disadvantages, which are related primarily to the lower carrying capacity, lower operating temperatures, relatively high-dimensional variations due to humidity conditions, etc. [11,12,13]. In order to improve the characteristics of polymer gears, some additives (such as glass, carbon, and aramid fibres) can be added with the purpose of increasing load capacity [14,15,16]. Furthermore, a temperature decrease between meshing gears can be achieved with lubricants such as Polytetrafluoroethylene (PTFE), graphite, or boron nitride [17,18].

When designing different machine components or engineering structures that contain polymer gears, the standardised procedure according to VDI 2736 [19] is usually used to estimate their load capacity against different types of failures: melting, tooth fracture, pitting, wear, and tooth deformation. Because polymer gears usually run in dry operating conditions (without lubrication), high contact friction and, consequently, a high degree of wear are often the main reasons for the appearance of critical failure in the analysed gear pair, especially in high-power-transmission applications [20]. However, the wear of highly loaded polymer gears may be reduced with applying low-frictional coatings [21], which may also improve the surface properties of teeth flanks [22,23]. Dearn et al. [24] investigated experimentally the influence of solid lubricant coatings (molybdenum disulphide-MoS2, graphite flake, boron nitride, and Poly-Tetra-Fluoro-Ethylene (PTFE)) on the wear behaviour of polymer gears. The authors concluded that the PTFE-coating provided the greatest reduction in wear for the analysed polymer gears. Furthermore, Bae et al. [25] and Petrov et al. [26] showed that very thin coatings (thinner than 2 μm) had a negligible effect on the contact stress of meshing polymer gears. Physical Vapour Deposition (PVD) is another technique that may be used to improve the wear resistance of contacting mechanical elements (PVD-coatings are described further in Section 2). As presented by Baptista et al. [27,28], Fereira et al. [29], Abdulah et al. [30], and Imbeni et al. [31], the PVD-technology has been used widely for the deposition of thin metal coatings on the polymeric substrate to improve wear resistance and other characteristics (optical enhancement, visual/aesthetical upgrading, etc.) of the analysed polymeric component. As already presented by the authors of this paper [1], the influence of very thin PVD-coatings (less than 500 nm) on the wear behaviour of POM polymer spur gears is small and does not reduce the wear significantly. For that reason, the multilayer (thicker) PVD-coatings made of Al, Cr, and CrN should be considered in the future. However, when using the metal PVD-coating on the basic polymeric component, the noise evaluation should also be taken into account.

Numerous research has been conducted in the past regarding the noise evaluation of metallic (especially steel) gears. On the other hand, a limited amount of research studies related to the noise evaluation of polymer gears may be found in the professional literature. Hoskins et al. [32] investigated the acoustic noise from polymer gears made of POM, PA66, or PEEK. In their study, it was concluded that the low noise as a typical characteristic of polymer gears does not account for the tribological noise generated as a result of the interacting tooth flanks. Sight et al. [33,34] investigated polymer spur gears with various functionally graded materials, influencing tribology properties and noise emission. According to their conclusions, the noise reduction was better with gears produced by injection moulding. Furthermore, the rotational speed was found as the most significant factor for the noise emission from the analysed polymeric gears. Nozowa et al. [35] studied the tribological properties of the nylon/steel gear pair and their influence on the noise emission. Authors concluded that noise emission was reduced by about 5 dB if compared with a steel/steel gear pair. Sharma et al. [36] investigated the noise and damping of polymer and composite spur gears operating at different rotational speeds. Their experimental results indicated that glass-fibre-reinforced polymer spur gears are more appropriate than metal gears in light-load-power-transmission applications due to their lower noise and damping factor. The vibration fault detection of polymer gears was studied by Kumar et al. [37]. Here, the statistical feature of the acquired signal was used for the identification of failure. Using this approach, the failure was detected successfully considering the high-order statistics indicators. Radionov et al. [38] investigated the acoustics characteristics of a gear pump with a polymer pinion shaft. The authors confirmed that polymer gears and shafts had a better acoustic feature than steel gears.

Based on the conclusions presented above, polymeric gears can be considered as being ‘low-noise’ components if compared to metal gears. This is due to the fact that their low modulus makes them resilient when teeth come into contact. However, polymeric gears are more sensitive to the wear of gear flanks, especially in the case of high loadings and high rotational speeds. To overcome this weakness, the appropriate metal coatings may be applied on the polymeric substrate to improve the wear resistance of polymer gears. However, the noise behaviour of a gear pair may change if coated polymer gears are used. To answer this question, comprehensive noise measurements and subsequent analyses of both uncoated and coated polymer gears made of POM were performed in this study. The PVD-coatings (Al, Cr, and CrN) used are described briefly in Section 2.2, while the testing procedure is discussed in Section 2.3.

## 2. Materials and Methods

### 2.1. Base Material

The polyoxymethylene (POM) with the material parameters shown in Table 1 was selected as a base material of the analysed polymer gears. The polymer gear specimens made of POM were machine-cut from extruded bars using a hobbing process. Some of the POM-gears were then coated with Al, Cr, or CrN PVD-coatings, as described in Section 2.2.

### 2.2. PVD-Coatings and Deposition Process

In the proposed experimental study, three different coatings were prepared on the polymer gears made of POM. The Aluminium (Al) coating was applied through a plasma activation process, followed by metallisation of the aluminium through a magnetron sputtering process, and, finally, a plasma polymerisation step. Chromium metallisation by the magnetron sputtering process was used for the Chromium (Cr) coating. The third coating of Chromium Nitride (CrN) was prepared in two steps, namely, the metallisation of chromium by the magnetron sputtering process, and, finally, the step of reactive metallisation of chromium and nitrogen by the magnetron sputtering process. All samples with all types of coatings were made-up on the same device, a META ROT 500 machine with a horizontal short cycle system for metallisation, including a part for spraying protective coatings.

Physical Vapor Deposition (PVD) is the process of applying thin, solid coatings to the base material by vaporisation. Coatings can be single-layer or multilayer. Multilayers can consist of different alloys [30,31]. They are dispersed at the atomic or molecular level. PVD is used in order to achieve high adhesion and hardness, to improve wear resistance, to improve tribological properties [27], and to improve optical properties [28,29] in various applications.

Several PVD techniques are known. The magnetron sputtering (MS) process with a prior plasma activation process and subsequent plasma polymerisation was used in this study. In the experimental work, we used plasma activation for the Al coating before the main MS process in order to improve the properties of the coatings on the polymer material. The POM polymer is known for its low surface energy and the plasma activation step improves the adhesion of the material. Plasma activation enables, in the first phase, the desorption of impurity molecules and the formation of radicals; in the second phase, the molecules disintegrate; in the third phase, the radicals of the gas molecules react on the surface of the polymer and, thus, increase its surface energy [40]. The entire process of plasma activation lasted 18 s with a regulation energy of 198 kWs.

Magnetron sputtering is a process where the material is vaporised by bombarding the target material with high-energy ions (Figure 1). The process takes place in a vacuum-sealed chamber, where there is an inert gas, substrate, and material for the coating [27]. A static magnetic field is formed in the chamber, which keeps the electrons close to the surface of the cathode, where the ionisation increases and forms a dense plasma. The plasma contains an ion with which the target is sputtered [41]. According to this principle, solid single or multi-layer coatings are prepared by magnetron sputtering. In the presented study, this process was used for all three coatings.

In the case of Al coating, plasma polymerisation was also used with the aim of improving the functional efficiency, i.e., the equipment of the Al coating on the base material in improving the mechanical properties of the polymer. In the second step of the CrN coating, reactive chromium + nitrogen metallisation was used according to the magnetron sputtering principle. In the reactive metallisation process, nitrogen gas was added to a vacuum chamber and made reactive by high-energy collisions. Thus, the nitrogen reacted chemically with the chromium target to create a molecular compound that formed a thin layer of chromium nitride. The process parameters of all three analysed coatings are shown in Table 2, while the appearance of coated POM-gears is shown in Figure 2.

### 2.3. Testing Procedure

The gears were tested on an in-house developed testing device, as shown in Figure 3. The testing device consists of two rigid steel blocks, which are connected firmly with two connecting bars; together, they form the rigid frame of the whole construction. The closed-loop consists primarily of two operating shafts connected with two gear pairs. During the experimental testing, the tested pinion made of POM was meshed with a support gear made of steel (the basic parameters of the gears are presented in Table 3). The rotational speed was set to *n* = 1000 rpm. A torque *T* = 20 Nm was applied with a plain digital torque wrench through the gear with a wrench gap at the accessories for working torque, which consisted of a one-way Clutch Bearing CSK 35 to avoid the back rotation of the shaft. Once the desired torque was applied, the clutch was closed and the tightening device could be removed.

The testing gears were protected with a sound-proof acoustic foam in order to decrease the noise caused by the electric motor, bearings, and toothed belt with pulley. Thus, the tested gears worked similar to a semi-anechoic chamber (see Figure 4).

The National Instruments NI PXI 4472 system and the AP 7046 microphone with PS9200 power supply were used to measure sound pressure. By increasing and reducing the distance of the microphone, the sound pressure level changes, and it is decreased with distance. Therefore, it is required to carry out all the measurements at a certain constant distance. The measurement of sound pressure level depends on several factors, i.e., on the source distance from the microphone, the direction of measurement, and the acoustics of the environment. However, the microphone distance can be reduced to a minimum, i.e., it can be very near the field of vibrating surfaces in order to avoid the problems of acoustic background. Getting very near the field of vibrating surfaces also depends on the mechanical source of vibrations and disturbances in the acoustic field between the source and the microphone. In our case, we placed the microphone at a distance of 100 mm, which was within the measuring space/housing protected with sound-proof acoustic foam, which is similar to a semi-anechoic chamber. The LabView software was used to analyse the signals that were then analysed in the time, frequency, and time–frequency domains by a program, which was developed on the basis of a LabView professional software version.

The quality of all gears had been checked before they were set on the testing device. Before measurement, the quality of all gears was checked with a coordinate measuring machine equipped with additional equipment for measuring gears and with the appropriate software. During assembly, the gears were carefully mounted on the shaft of the testing device and operated to the appropriate operating condition. All tests were made in the representative number of measuring tests. The Sound Pressure Level (SPL) is the sound pressure expressed in decibels (dB).

The sound signal was obtained in the phase of normal gear operation in the testing device. The sound was measured in the running-in phase and in the wear-in phase of the life cycle during increased surface degradation and wear of coatings of a gear pair. Thus, the condition was monitored by measuring the sound during the entire test. After the running-in phase (approximately 4 min), patterns of spectra and spectrograms already appeared at the beginning of the wear-in phase of the life cycle. As we were interested in the sound difference between coated polymer gears and those without coating, we observed coating surfaces. The criteria were stability and the presence of coatings in the tooth flanks. We finished with measurements when the peeling of the coating was more than 80%. The test duration was up to 30 min.

### 2.4. Frequency Analysis

For application, a Discrete Fourier Transform (DFT) is necessary. In general, perfect gear sound signals are periodical. Harmonics of the meshing frequency are also included in the signal spectrum due to non-linearities in the meshing process. However, gears are, in reality, never perfect. As teeth spacing is usually not a perfect constant, the contact point oscillates. In addition, the coating is peeled, which causes additional hindrances on the surface of gear flanks that are in contact, i.e., disturbances in the meshing process occur.

### 2.5. Time–Frequency Analysis

In concern to signals used for technical analysis, some frequencies appear only in some cases. When using classical frequency analysis of such signals, the time when particular frequencies appear in the spectrum cannot be determined. Time–Frequency Analysis (TFA) is used in order to determine how frequencies of nonstationary signals change with the time and how intense they are [42].

For Short Time Fourier Transformation (STFT), a time signal is divided into short time intervals and, afterward, frequency analysis of each interval is carried out separately. STFT is a linear time–frequency transformation. In order to eliminate the defects of Fourier transformation, signals are compared with elementary functions, determined in time space and in frequency space.

The Fourier transform of the signal *x*(*t*) does not suffice for the frequency domain analysis if it is non-stationary. It is necessary to divide the signal into segments before performing the Fourier analysis. Assumably, the signal is stationary within each segment. Such a signal divided into segments is called a windowed signal:(1)$$x_{w}\left(t\right)=x\left(t\right)\cdot w\left(t\right)$$

For such a windowed signal, the result of the Fourier transform can be called a windowed Fourier transform because it is a function of frequency and windows position:(2)$$\mathrm{STFT}\left(f,\tau \right)=X_{w}\left(f,\tau \right)=\int_{-\infty }^{\infty }x\left(t\right)\cdot w\left(t-\tau \right)e^{-j2\pi ft}dt$$

The selection of the window function *w*(*t*) is possible in such a way that its Fourier transform *W*(*f*) is also a window function. The windowed Fourier transform presented in Equation (2) is often called STFT. In engineering applications, the square of the modulus of STFT is called a spectrogram. For each position of the window, it is possible to acquire different spectra. The total number of these spectra is a function that represents a time–frequency distribution.

## 3. Results and Discussion

Generally, noise is caused by the meshing process between the teeth of the gear. Teeth flanks slide and roll during the rotational motion of gears. At the same time, owing to load, teeth deflection appears due to stiffness changes within the mesh. Vibration is produced due to dynamic behaviour of teeth deflection, and this is a primary source of the noise. Additional noise appears due to geometrical errors and different failures of gears, e.g., due to the peeling of various types of coatings of polymer gears. An aero-acoustical type of noise is directly generated gear noise.

The microphones were calibrated before starting the noise measurements. As sound measurement was relatively accurate, the appropriate sound pressure level could be determined without distortion. The manufacturer’s recommendations regarding the distance and direction of orientation were followed when using the microphone. The operation of electric motors by means of frequency converters was carried out in such a way to ensure the smallest electroacoustic disturbances in reference to the design of the test device. The soundproof foam was applied in order to separate the measuring space and the surroundings.

Afterward, the noise of the operating gears was measured as a sound signal. In Figure 5, Figure 6, Figure 7 and Figure 8, time signals of the sound pressure are shown. The amplitude of the sound signal of polymer gears without coatings is smallest (Figure 5). Polymer gears with Al-coatings (Figure 6) have a slightly larger amplitude, followed by polymer gears with CrN-coatings (Figure 8). In the case of polymer gears with Cr-coatings (Figure 7), the amplitude sound pressure is largest. In case of polymer gears with a 3-layer coating, the sound pressure is smaller than in the case of polymer gears with 5-layer coatings. With the increased number of layers, the sound level pressure increases as well. Thus, the sound pressure of polymer gears with 3-layer coatings is lower than the sound pressure of polymer gears with 5-layer coatings.

Figure 9 presents the sound pressure levels *L* and *L*_max_ of polymer gears with or without different coatings for different coating layers. Sound pressure measured in Pa is the dynamic variation in the static pressure of air. In order to obtain a sound pressure level, the instantaneous sound pressure is averaged over a certain duration. The sound pressure level is usually represented on a logarithmic amplitude scale, similar to the human perception of hearing.

The sound signal was obtained in the phase of normal gear operation in the testing device. The above values apply for the running-in phase and wear-in phase in the life cycle of a gear pair. The amplitude of the sound pressure level *L* of polymer gears without coatings is lower than the amplitude of polymer gears with coatings. In view of the reference value of sound pressure level 89.1 dB for polymer gears without coatings, the sound pressure level of polymer gears with Al-coatings is higher by 0.3 dB, whereas the sound pressure level of polymer gears with Cr-coatings is higher by 1.2 dB, and for polymer gears with CrN-coatings, it is higher by 0.4 dB. All this applies for coatings with 3 layers. In the case of coatings with 5 layers, the sound pressure level of polymer gears with Al-coatings is higher by 0.6 dB, whereas the sound pressure level of polymer gears with Cr-coatings is higher by 1.8 dB, and when it comes to polymer gears with CrN-coatings, it is higher by 0.7 dB.

When comparing the maximum or peak sound pressure level *L*_max_ in the phase of the life cycle of increased surface degradation and wear of coatings, the amplitude of the sound pressure level of polymer gears without coatings is lower than the amplitude of polymer gears with coatings. If the maximal sound pressure level of polymer gears without coatings has the reference of 94.7 dB, the sound pressure level of polymer gears with Al-coatings is higher by 0.9 dB, whereas the maximum sound pressure level of polymer gears with Cr-coatings is higher by 2 dB, and when it comes to polymer gears with CrN-coatings, the maximum sound pressure level is higher by 1 dB. All this applies for coatings with 3 layers. In the case of coatings with 5 layers, the sound pressure level of polyamide gears with Al-coatings is higher by 1.1 dB, whereas the sound pressure level of polyamide gears with Cr-coatings is higher by 2.7 dB, and when it comes to polyamide gears with CrN-coatings, it is higher by 1.2 dB.

When comparing the maximum sound pressure level, polymer gears without coatings have a lower amplitude. In the running-in phase, the sound pressure level is slightly higher than in the wear-in phase in the life cycle. The increase in the operating temperature leads to a slight reduction in the noise in the wear-in phase. Afterward, the sound pressure level increases significantly (by more than 4 dB) in the phase of the life cycle of increased surface degradation, and the termination of coating is complete.

In relation to acoustic measurements, the final sensor is often the human ear, meaning that acoustic measurements often try to describe the subjective perception of a sound by the human ear. Devices usually provide a linear response. On the other hand, the ear is a nonlinear sensor. Consequently, filters, referred to as psophometric weighting filters, are applied to account for the nonlinearities. In Table 4, sound pressure levels are presented with different A-, B-, and C-weighting filters.

In the frequency analysis, a measured signal with a stable rotational speed must be ensured. Thus, it is required to monitor the rotational speed of gear pairs. If the rotational speed deviates, this leads to some problems in relation to frequency analysis. In the test, the maximum deviation of the rotational speed is 0.38%, which does not significantly impact the frequency analysis.

The measured sound signal was processed with Hamming windows and analysed with the fast frequency transform, without filters. The sampling rate for obtaining the signal was 65,536 samples/s. In Figure 10, Figure 11, Figure 12 and Figure 13, the frequency spectrum of the sound signals of polymer gears with and without coatings with different coating layers is presented. The meshing frequency and high harmonics are also noted. Additionally, sidebands around the previously mentioned typical frequencies (600 Hz, 1200 Hz, and 1800 Hz) are observed.

When comparing the frequency spectra, the amplitudes of dominated frequency components of polymer gears without coatings are lower (Figure 10). The sidebands around the meshing frequency and high harmonics components increase significantly in the case of polymer gears with coatings. Some added frequency components can be observed for polymer gears with coatings in the frequency spectrum with sidebands around the first three harmonics. The intensity of dominated frequency components increases with layers.

The dominated frequency components and their harmonics have many sidebands, particularly polymers gears with different coatings. These sidebands are a semi-periodical phenomenon connected with tooth flanks. As the roughness of tooth flanks is directly connected with friction, more side asymmetrical frequency components are produced in frequency spectra. On the tooth flank with a coating, the coating degraded and peeled during operation. Parts where the coating was peeled caused a local increase in friction and different intensity of the slip-stick effect. When observing the intensity of amplitude of polymer gears with coatings, a minor increase in the amplitude of polymer gears with Al-coatings (Figure 11) and CrN-coatings (Figure 12) can be noted. The spectra for polymers gears with Cr-coatings (Figure 13) have a significantly increased amplitude.

In the case of the time–frequency analysis, the measured signal was 2 s long, the frequency sampling was 65,536 samples/s, and the window length was 80 ms. The measured sound signal was processed with Hanning windows and analysed with fast frequency transformation without filters.

When the frequency spectrogram is analysed, the typical frequency components are observed. It is possible to observe how frequency changes during the time. In Figure 14, the frequency spectrogram of the sound signal of the polymer gears without coatings is presented. In the time space at low frequency, low stochastic changes in the frequency amplitude can be noted, and for frequency components at 1800 Hz and 2400 Hz with a low amplitude, partial quasi-periodical changes with time can be observed.

In Figure 15, the frequency spectrogram of a sound signal of the polymer gears with Al-coatings is presented. At the second and third harmonics (1800 Hz and 2400 Hz), it is possible to notice partial periodical pulsation of the frequency components for 3-layer coatings. In 5-layer Al-coatings, the pulsation increases and, additionally, first harmonics can be noticed at 1200 Hz. Added layers increase pulsation.

The frequency spectrogram of a sound signal of the polymer gears with Cr-coatings is presented in Figure 16. At the third and fourth harmonics, periodical pulsation of the frequency components can be noted for 3-layer coatings. In 5-layer Cr-coatings, it is possible to notice intensive pulsating at the second, third, and fourth harmonics, and for the meshing frequency at 600 Hz, intensive pulsating with a stochastic energy distribution inside the pulse.

In Figure 17, the frequency spectrogram of a sound signal of the polymer gears with CrN-coatings is presented. In 3-layer coatings, a new phenomenon is noticed. At the third harmonics (2400 Hz), periodical pulsation of the frequency components with energy redistribution or dual uneven redistribution can be noticed. The meshing frequency of 600 Hz and first harmonic of 1200 Hz pulsated with a stochastic energy distribution inside the pulse. In 5-layer CrN-coatings at the third harmonics (2400 Hz), pulsation with a double-frequency distribution can be noticed. At meshing frequency 600 Hz, there is pulsation with an energy stochastics redistribution.

Analysis in the time–frequency domain is a powerful tool for identifying the presence of pulsation for typical frequency components. Typical intensities and classification of pulsation are connected with friction because the slip-stick effect or moving and stopping the tooth flank on the surface contact is dominant. As the wear resistance and durability of specific coatings are different, this leads to peeling with different phenomena. Peeling parts with specific shape, size, and strength cause hacking into the teeth surface, accumulation, and sticking into the contact meshing point of the teeth flank. The degradation of the coated surface with peeling for polymer gears with 3-layer coatings after 22 min is presented in Figure 18, and that for polymer gears with 5-layer coatings is presented in Figure 19. It is evident that the surface degraded with peeling more than 90% in the case of polymer gears with 3-layer coatings and more than 85% in the case of polymer gears with 5-layer coatings.

## 4. Conclusions

This paper presents an experimental study on the noise emission analysis of polymer gears made of POM and coated with Al-, Cr-, and CrN- multilayer PVD-coatings (surface coatings with 3 and 5 layers were considered). Standard cutting tools were used to machine involute gears, and PVD-coatings were produced using a special deposition process. The gear pairs were tested and analysed using a special custom-made testing device. The results for polymer gears with coatings obtained during the experiments were compared with the results of polymer gears without coatings.

Based on the theoretical study and obtained experimental results, the following conclusions can be drawn:
The sound pressure level of polymer gears without coatings is lower than that of polymer gears with different coatings. The increase in sound pressure level is 0.8 dB on average. As these coatings are moderately resistant, they are peeled during operation, and parts of the coating come into the contact point of the meshing area and cause friction and disturbances in the meshing process.With the increased number of layers, the sound pressure level increases as well, as the quantity of coating peeling parts in the meshing points is larger. The increase in sound level pressure is from 20% to 60% for various coatings.When using softer coatings, such as the Al-coating, the increase in the sound pressure level is smaller as the coating is softer and, in the peeling phase, the parts are smaller and they are crushed easier when they enter the contact point of the meshing area. In the case of harder coatings, such as Cr-coatings and CrN-coatings, the peeled-off parts are significantly harder; they do not crush easily and are kneaded into the basic material more intensely and accumulate in the contact point of the meshing area. Thus, they increase local friction, disturbances in the meshing process, and noise emission.With the time–frequency method, it is possible to observe in the acoustics signal how coatings on the tooth flank are degraded and peeled off and which phenomena that identify energy distribution are present in the sound signal. The technique is useful for identifying the pulsation, which causes changes in the friction and slip-stick effect in the contact point in the meshing area with the change in time.

## Figures and Tables

**Figure 1 polymers-15-00783-f001:**
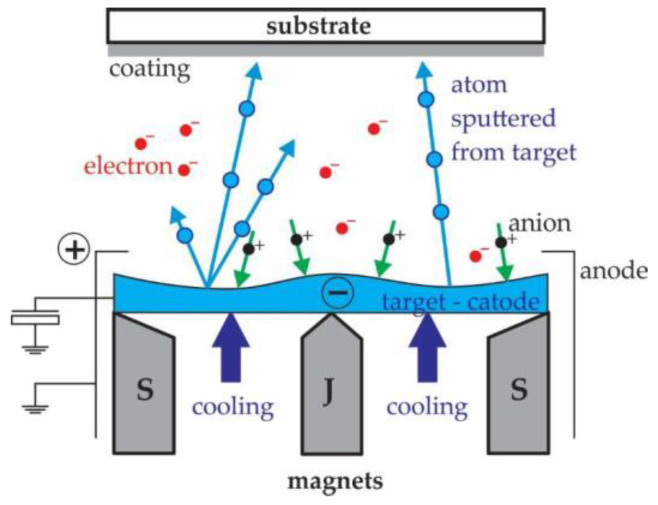
Magnetron sputtering process [1].

**Figure 2 polymers-15-00783-f002:**
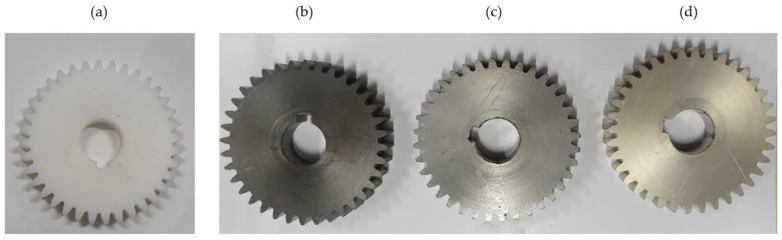
Tested gears made of POM: (**a**) Without coating, (**b**) CrN-coating, (**c**) Al-coating, (**d**) Cr-coating.

**Figure 3 polymers-15-00783-f003:**
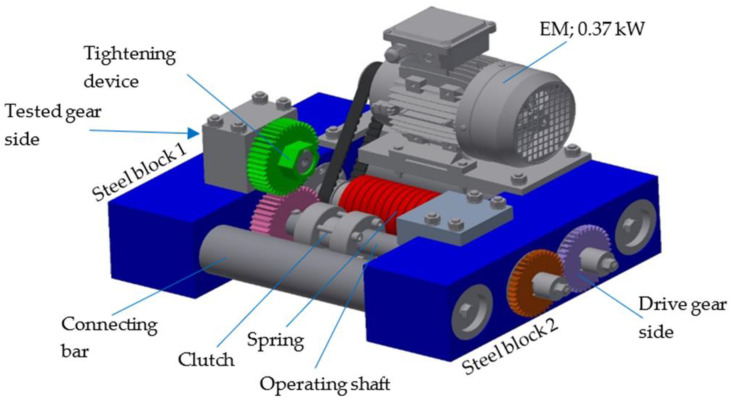
The principle of the testing device.

**Figure 4 polymers-15-00783-f004:**
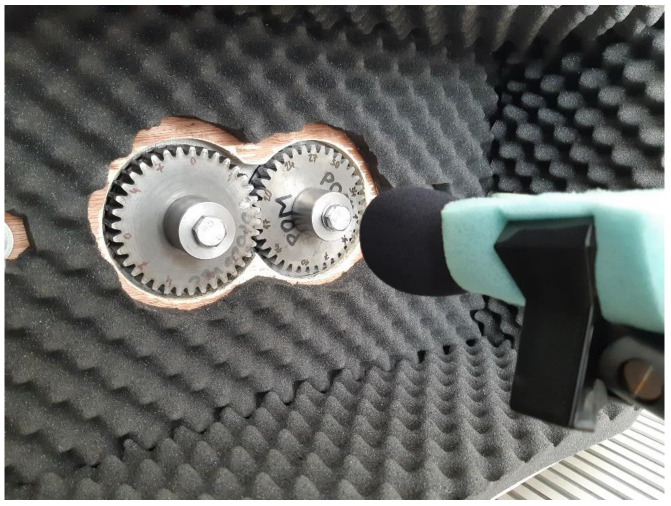
The tested gear pair on the testing device with the soundproof acoustic foam for acoustic soundproof insulation.

**Figure 5 polymers-15-00783-f005:**
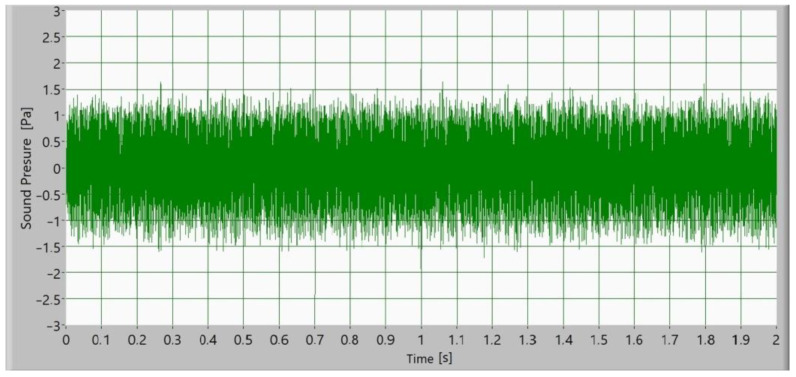
Time signal of sound pressure for polymer gears without coating.

**Figure 6 polymers-15-00783-f006:**
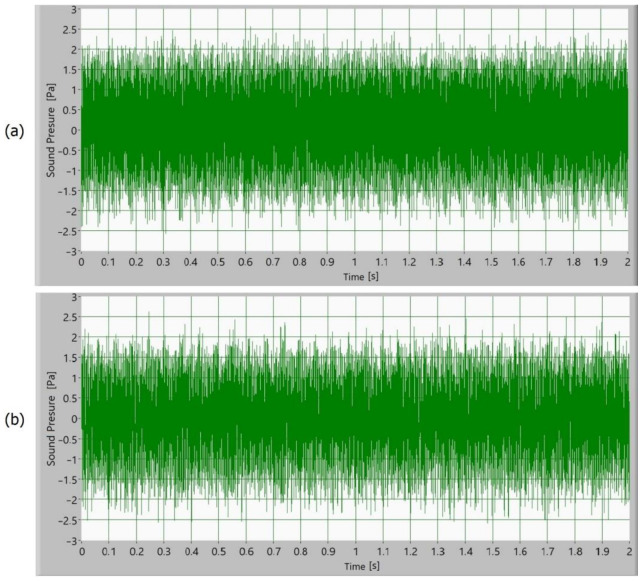
Time signal of sound pressure for polymer gears with Al-coating with (**a**) 3-layer coating and (**b**) 5-layer coating.

**Figure 7 polymers-15-00783-f007:**
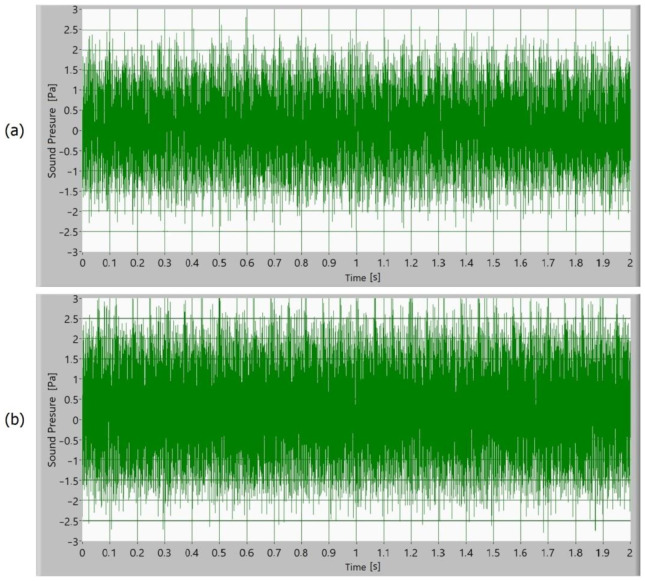
Time signal of sound pressure for polymer gears with Cr-coating with (**a**) 3-layer coating and (**b**) 5-layer coating.

**Figure 8 polymers-15-00783-f008:**
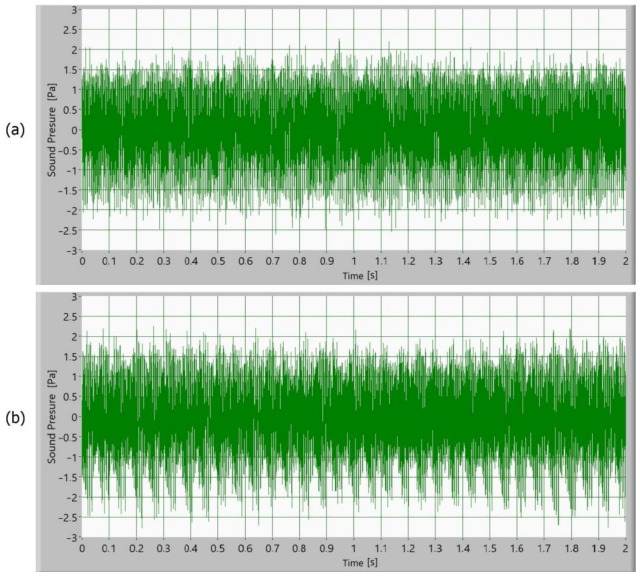
Time signal of sound pressure for polymer gears with CrN-coating with (**a**) 3-layer coating and (**b**) 5-layer coating.

**Figure 9 polymers-15-00783-f009:**
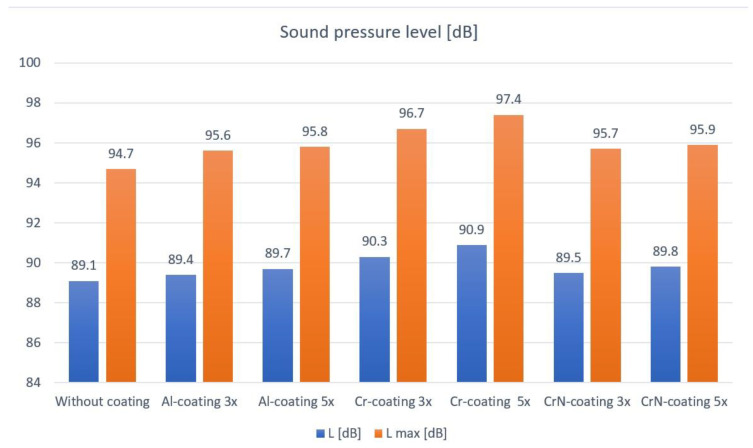
Sound pressure levels L and L_max_ of polymer gears with and without coating.

**Figure 10 polymers-15-00783-f010:**
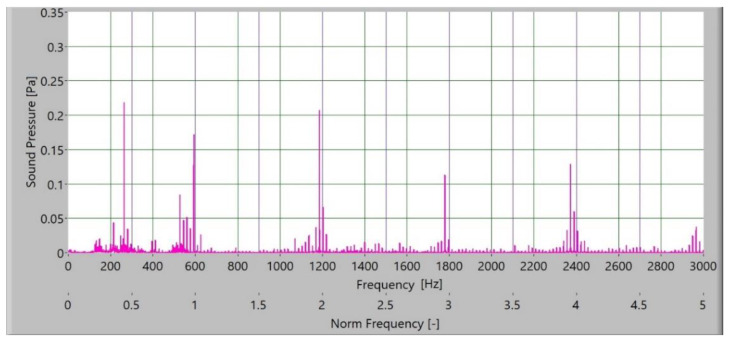
Frequency spectrum of sound pressure for polymer gears without coating.

**Figure 11 polymers-15-00783-f011:**
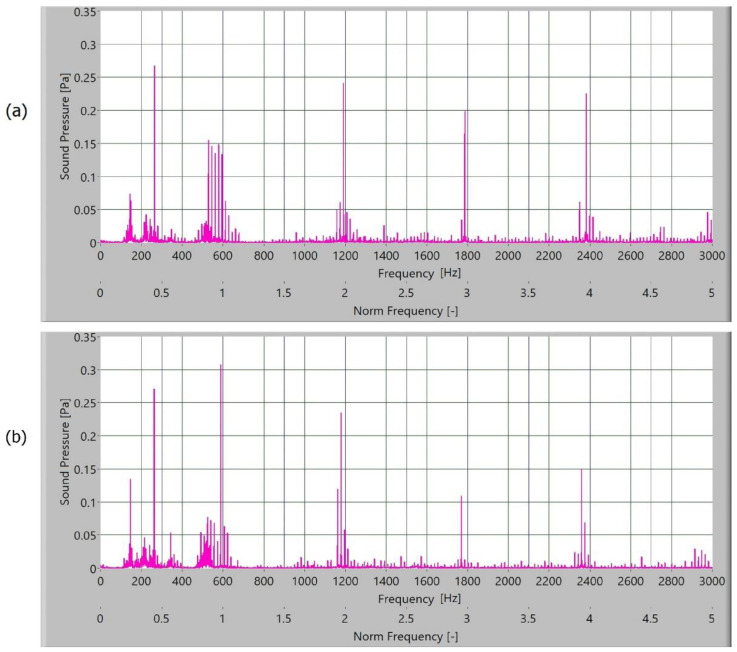
Frequency spectrum of sound pressure for polymer gears with Al-coating, with (**a**) 3-layer coating and (**b**) 5-layer coating.

**Figure 12 polymers-15-00783-f012:**
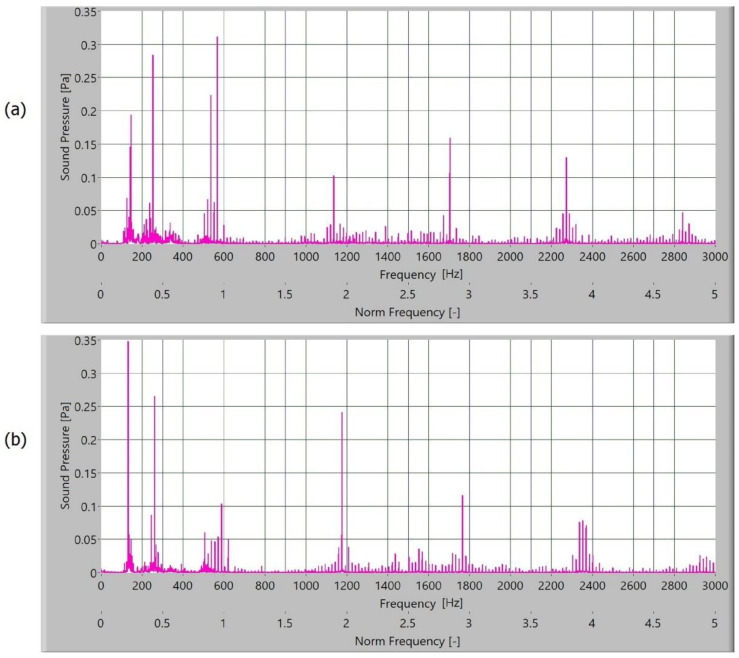
Frequency spectrum of sound pressure for polymer gears with Cr-coating, with (**a**) 3-layer coating and (**b**) 5-layer coating.

**Figure 13 polymers-15-00783-f013:**
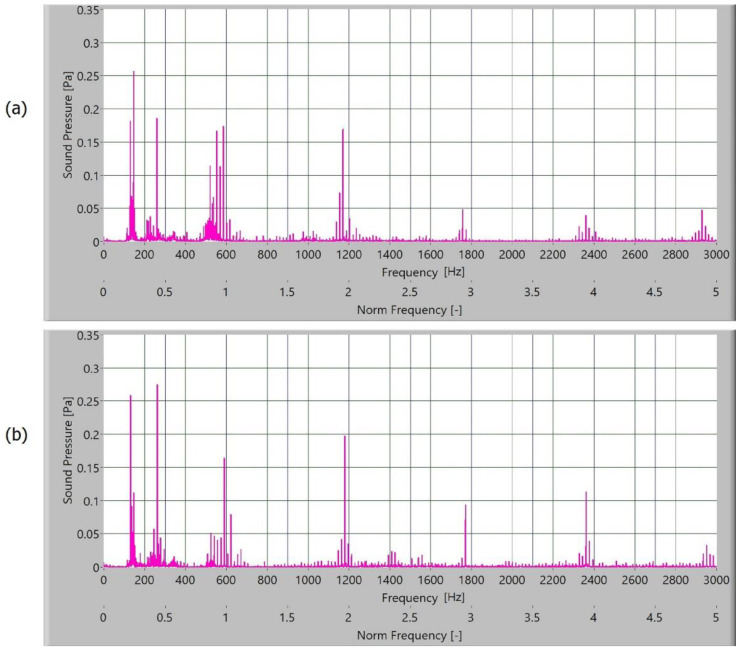
Frequency spectrum of sound pressure for polymer gears with CrN-coating, with (**a**) 3-layer coating and (**b**) 5-layer coating.

**Figure 14 polymers-15-00783-f014:**
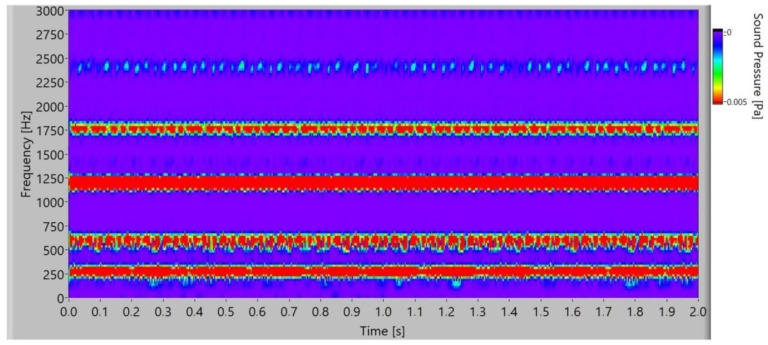
Frequency spectrogram of sound signal for polymer gears without coating.

**Figure 15 polymers-15-00783-f015:**
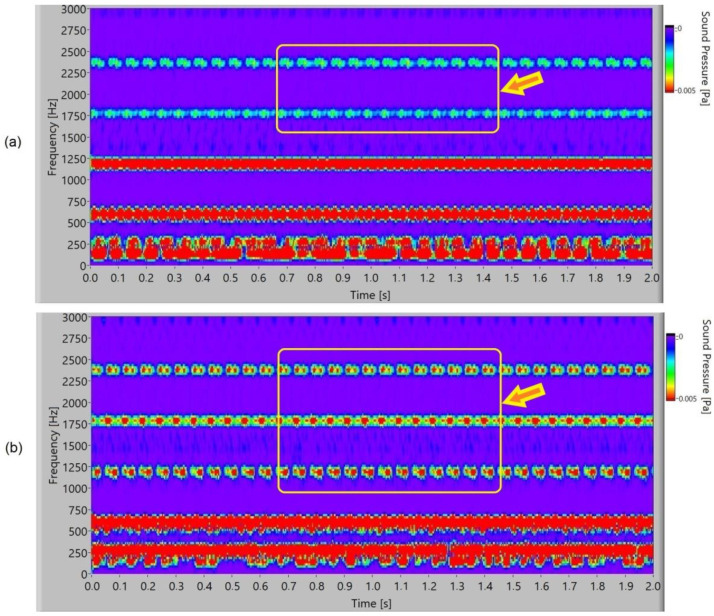
Frequency spectrogram of sound signal for polymer gears with Al-coating: (**a**) 3 layers; (**b**) 5 layers.

**Figure 16 polymers-15-00783-f016:**
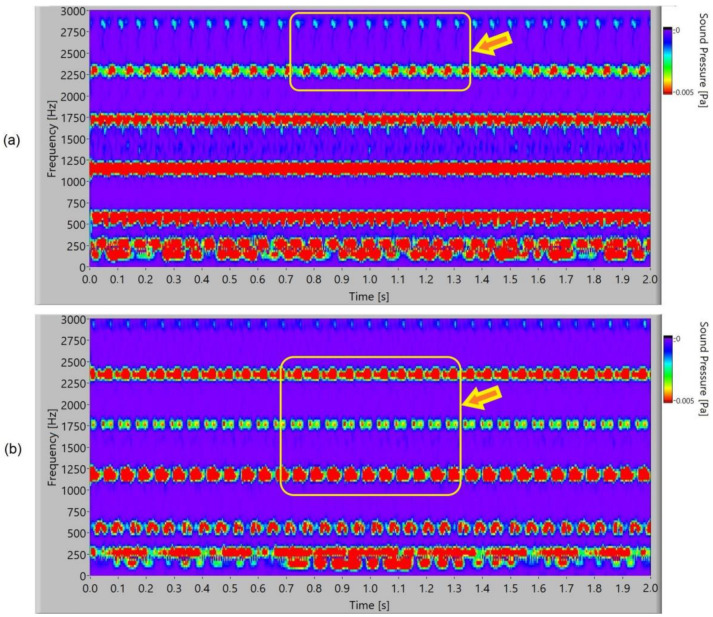
Frequency spectrogram of sound signal for polymer gears with Cr-coating: (**a**) 3 layers; (**b**) 5 layers.

**Figure 17 polymers-15-00783-f017:**
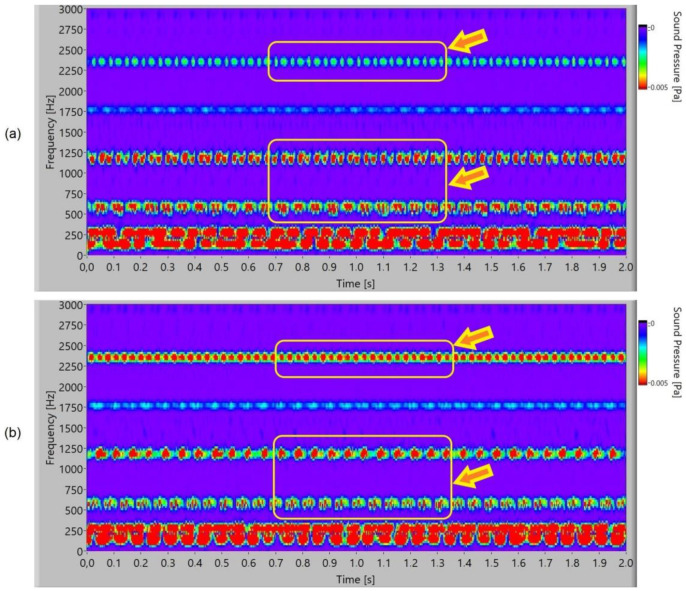
Frequency spectrogram of sound signal for polymer gears with CrN-coating: (**a**) 3 layers; (**b**) 5 layers.

**Figure 18 polymers-15-00783-f018:**
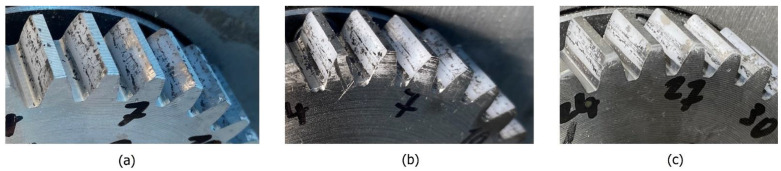
Peeling of polymer gear flanks with 3-layer coatings: (**a**) Al, (**b**) Cr, (**c**) CrN.

**Figure 19 polymers-15-00783-f019:**
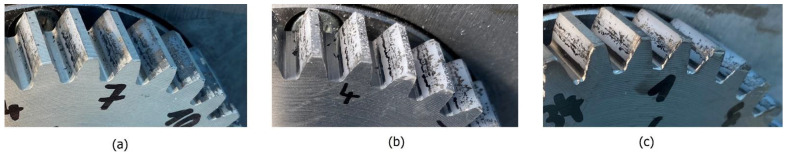
Peeling of polymer gears flanks with 5-layer coatings: (**a**) Al, (**b**) Cr, (**c**) CrN.

**Table 1 polymers-15-00783-t001:** Material parameters of the base material (POM) [39].

Mechanical Characteristics	Standard	Unit	Value
Yield stress (+23 °C, dry)	ISO 527-1/-2 DIN 53455 ASTM D 638	MPa (N/mm^2^)	67
Tensile strength (+23 °C, dry)	ISO 527-1/-2 DIN 53455 ASTM D 638	MPa (N/mm^2^)	66
Elongation at break (+23 °C, dry)	ISO 527-1/-2 DIN 53455 ASTM D 638	%	40
Tensile E-modulus (+23 °C, dry)	ISO 527-1/-2 DIN 53455 ASTM D 638	MPa (N/mm^2^)	2800
Charpy notched impact strength (+23 °C, dry)	ISO 179 DIN 53453	kJ/m^2^	6
Charpy notched impact strength (+23 °C, dry)	ISO 179/1eA	kJ/m^2^	8
Ball indentation hardness (dry)	ISO 2039-1	MPa (N/mm^2^)	130
Thermal characteristics	Standard	Unit	Value
min. Operating temperature (continuous)	-	°C	−50
max. service temperature (continuous)	-	°C	100
max. service temperature (short-term)	-	°C	140
Heat Deflection Temperature HDT/A (1.8 N/mm^2^)	ISO 75-1/-2 DIN 53461 ASTM D 648	°C	100
Thermal conductivity (+23 °C)	DIN 52612	W/(m × K)	0.31
Combustibility characteristics	Standard	Unit	Value
UL94 flammability	IEC 60695-11-10	class	HB
Electrical characteristics	Standard	Unit	Value
Dielectric constant, relative permittivity (1 MHz, dry)	DIN IEC 60250 (DIN VDE 0303-4) ASTM D 150		3.8
Dielectric loss factor (1 MHz, dry)	DIN IEC 60250 (DIN VDE 0303-4) ASTM D 150		0.005
Surface resistivity (dry)	DIN IEC 60093 (DIN VDE 0303-30) ASTM D 257	Ω	10^13^
Physical characteristics	Standard	Unit	Value
Density	ISO 1183 DIN 53479 ASTM D 792	g/cm^3^	1.41
Moisture absorption at saturation (23 °C /50%r.h.)	ISO 62 ISO 1110	%	0.20
Water absorption at saturation (water storage 23 °C)	ISO 62 DIN 53495 ASTM D 570	%	0.8

**Table 2 polymers-15-00783-t002:** Process parameters of the analysed PVD-coatings.

Coating	Process	Pumping Time (s)	Staring Pressure (mbar)	Mass Flow Contr. MFC	Regulation Pressure (mbar)	Process Time (s)	Regulation Energy (kWs)	*T* (°C)	
								Min	Max
Al	Plasma activation	10	5⋅10^−3^	800	3⋅10^−2^	18	198	500	5000
	Magnetron sputtering	150	4⋅10^−4^	500	2.2⋅10^−3^	62	10,500	30	90
	Plasma polymerisation	1	1.5⋅10^−2^	300	2⋅10^−2^	50	582	500	5000
Cr	Magnetron sputtering	80	6⋅10^−4^	500	3⋅10^−3^	105	10,200	25	90
CrN	Magnetron sputtering	80	6⋅10^−4^	500	3⋅10^−3^	105	10,200	25	90
	Reactive metallisation	90	9⋅10^−4^	120 (190)	3.4⋅10^−3^	67	6200	40	90

**Table 3 polymers-15-00783-t003:** Basic parameters of the tested gear pair.

Parameter	Tested Gear	Supported Gear
Material	POM	Steel (16MnCr5)
Normal module *m*	2.5 mm	2.5 mm
Pressure angle α_n_	20°	
Helix angle β	0°	
Number of teeth *z*	36	36
Tooth width *b*	14 mm	14 mm
Profile shift coefficient *x*	0	
Centre distance a	90 mm	
Basic rack profile	ISO 53 A	
Lubrication	Dry (not lubricated)	

**Table 4 polymers-15-00783-t004:** Sound pressure levels in linear and A-, B-, and C-weighting filters of polymer gears with and without coating.

SPL (dB)	L	L_max_	L_(A)_	L_(A)max_	L_(B)_	L_(B)max_	L_(C)_	L_(C)max_
Without coating	89.1	94.7	88.2	94.2	88.5	94.2	88.8	94.4
Al-coating 3×	89.4	95.6	88.4	95	88.7	95.1	89.1	95.2
Al-coating 5×	89.7	95.8	88.7	95.2	89	95.3	89.3	95.3
Cr-coating 3×	90.3	96.7	89.5	96.3	89.7	96.4	90	96.4
Cr-coating 5×	90.9	97.4	90	96.9	90.3	97	90.7	97.1
CrN-coating 3×	89.5	95.7	88.6	95.2	88.8	95.4	89.2	95.3
CrN-coating 5×	89.8	95.9	88.9	95.3	89.2	95.5	89.6	95.5

## Data Availability

The data presented in this study are available on request from the corresponding author.

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
