# Peer review of "Noise Evaluation of Coated Polymer Gears"

_polymers, 2023, doi:10.3390/polym15030783_

Round 1
Reviewer 1 Report
Paper is a good example of practical results that are clearly showing difference of noise for differente coated polimer gears.
For producers of gears would be very interesting to know about the price. How much is coating increasing the price of single gear. Since you are having more coatings, it would be good to have some kind of comparison.
Conclusion is longer than what I would expect since the results are clear. Last bullet is conclusion of the fist three bullets and it should be deleted.
Bullet before tha last has similiar problem and it should be also deleted.
Conclusions need to be expanded like this: "The sound pressure level of polymer gears without coating is lower than that of polymer gears with different coatings. " This is clear and logical but I would expect here exact data like in percentige or in dB. So, how much lower is the sound pressure of polimer gear without coating comapring with gears with coating.
Author Response
Comment #1: For producers of gears would be very interesting to know about the price. How much is coating increasing the price of single gear. Since you are having more coatings, it would be good to have some kind of comparison.
Response: All samples with all types of coatings were made-up on the same device, a META ROT 500 machine with a horizontal short cycle system for metallization, including a part for spraying protective coatings. At this point, it is impossible to give the price of a single gear with a specific coating, because it should be necessary to do the entire evaluation process for an individual coating, which was not the subject of this investigation.
Comment #2: Conclusion is longer than what I would expect since the results are clear. Last bullet is conclusion of the first three bullets and it should be deleted.
Response: It has been corrected as suggested.
Comment #3: Bullet before the last has similiar problem and it should be also deleted.
Response: It has been corrected as suggested.
Comment #4: Conclusions need to be expanded like this: "The sound pressure level of polymer gears without coating is lower than that of polymer gears with different coatings. " This is clear and logical but I would expect here exact data like in percentige or in dB. So, how much lower is the sound pressure of polymer gear without coating comparing with gears with coating.
Response: The explanation has been added in Section 3 (Results and discussion).
Reviewer 2 Report
The article is well written. State-of-the-art, experimental set-up and novelty are well described. The article is of scientific and technical interest because plastic gear, coatings and gear noise are interesting topics for the gear scientific community.
However:
· Data are not sown in a clear way. Diagrams seem like screenshots of the authors' data elaboration software. Data visualization must be improved. The diagrams presented here are almost impossible to read.
· Also, as the authors are dealing with meshing gears, it would be interesting to normalize the frequency and the time scale based on the rotational frequency (or the 1st meshing frequency) and the number of rotations.
· In order to be able to compare the data, all of the diagrams should have the same scale.
· ISO 53 does not indicate a unique basic rack profile. Please clarify addendum, dedendum and root radius coefficients.
· ISO 3746:2010 suggest using data in terms of sound pressure level rather than simple sound pressure. Indeed, we humans are sensitive to the level (i.e. the log) of the sound rather than its value on a linear scale.
· Why only one rotational and torque level have been selected?
· Was the gear quality kept the same for all test cases? If not, this affects the results as high-quality gears have a different sound behaviour than low-quality ones
I would also like to ask to the authors if, all their reasoning about the peeling (i.e. the first points of the conclusions) is supported by the experimental evidence and if so, why physical evidence (e.g. pictures) of the physical phenomena have not been shown. Furthermore, if this was related to peeling, why it is not a continuous phenomenon? Instead, it seems more like a phenomenon occurring every cycle. On this point, I would also point out that, if peeling is already affecting the gear pair behaviour in a 2-second test, this is a big problem. Indeed, I expect that those gears were designed to last more than a few minutes…
I would also ask the authors if there were a reason for their design of the gears. Typically, gear pairs are designed with a coprime number of teeth (like 35 and 36) in order to redistribute the contact on all teeth. Instead, with a number that is not coprime, one pinion tooth will be in contact only with a certain group of wheel teeth. If the number of teeth is the same, like in the article, each pinion tooth will mesh only with a single wheel tooth. This aspect will also affect the effect of misalignments (and similar) on the dynamic behaviour. I think that this aspect is one affecting the repetitive behaviour highlighted by the frequency spectrogram. Indeed, the phenomenon is occurring at each rotational cycle.

Author Response
Comment #1: Data are not sown in a clear way. Diagrams seem like screenshots of the authors' data elaboration software. Data visualization must be improved. The diagrams presented here are almost impossible to read.
Response: The quality of the Figures and Diagrams has been improved as suggested.
Comment #2: Also, as the authors are dealing with meshing gears, it would be interesting to normalize the frequency and the time scale based on the rotational frequency (or the 1st meshing frequency) and the number of rotations.
Response: All Figures with frequency spectrum have been checked, and their quality has been improved. Furthermore, the Norm Frequency axis has been added in the revised manuscript.
Comment #3: In order to be able to compare the data, all of the diagrams should have the same scale.
Response: The uniform scale was used in the revised manuscript as suggested.
Comment #4: ISO 53 does not indicate a unique basic rack profile. Please clarify addendum, dedendum and root radius coefficients.
Response: We used the standard ISO 53 A: Pressure angle, αn = 20°; Addendum coefficient, ha∗ = 1.00; Dedendum coefficient, hf* = 1.25; Root radius coefficient, ρF∗ = 0.38.
Comment #5: ISO 3746:2010 suggest using data in terms of sound pressure level rather than simple sound pressure. Indeed, we humans are sensitive to the level (i.e. the log) of the sound rather than its value on a linear scale.
Response: For technical acoustics and diagnostics, we presented the results in the form of sound pressure on a linear scale; for human perception, we also added the Sound pressure level in dB with different filters (A, B, etc.) for all types of polymer gears with different coatings.
Comment #6: Why only one rotational and torque level have been selected.
Response: We used typical operating conditions (rotational frequency and torque) for special industrial applications. We were also limited with the number of polymer gears with coating as we gave priority to the correct measurement process and adequate measuring results.
Comment #7: Was the gear quality kept the same for all test cases? If not, this affects the results as high-quality gears have a different sound behaviour than low-quality ones.
Response: The quality of all gears was checked, and the same quality level was kept for all test cases.
Comment #8: I would also like to ask to the authors if, all their reasoning about the peeling (i.e. the first points of the conclusions) is supported by the experimental evidence and if so, why physical evidence (e.g. pictures) of the physical phenomena have not been shown. Furthermore, if this was related to peeling, why it is not a continuous phenomenon? Instead, it seems more like a phenomenon occurring every cycle. On this point, I would also point out that, if peeling is already affecting the gear pair behaviour in a 2-second test, this is a big problem. Indeed, I expect that those gears were designed to last more than a few minutes….
Response: Photographs of the peeling of coated polymer gears with the appropriate explanations have been added in the revised version of the manuscript.
Comment #9: I would also ask the authors if there were a reason for their design of the gears. Typically, gear pairs are designed with a coprime number of teeth (like 35 and 36) in order to redistribute the contact on all teeth. Instead, with a number that is not coprime, one pinion tooth will be in contact only with a certain group of wheel teeth. If the number of teeth is the same, like in the article, each pinion tooth will mesh only with a single wheel tooth. This aspect will also affect the effect of misalignments (and similar) on the dynamic behaviour. I think that this aspect is one affecting the repetitive behaviour highlighted by the frequency spectrogram. Indeed, the phenomenon is occurring at each rotational cycle.….
Response: In the proposed research, we used the existing testing device with the transmission rate 1 on the both sides (drive gear side and tested gear side). The Authors agree with the Reviewer that it is not the optimal solution because each pinion tooth will mesh only with a single-wheel tooth. However, the Reviewer's suggestion could be considered in our further research.
Reviewer 3 Report
In this study, noise evaluation of coated polymer gears investigated. The work is interesting and publishable. The comments of the referee about the study are given in below:
1. The pictures in Figure 2 are not clear (especially b and c). Can you add clearer photos?
2. Section 2.3 shows the test conditions. In this section, you can also specify the test duration.
3. In section 2.3 it is said that the microphone is closer to 100 mm from the test gears. Have you looked at what change happens when it's closer or farther away? If you have, can you briefly explain? How did you determine the 100 mm distance? Did you measure the sound level of the testing device before the gears were running?
4. In line 186 authors said “The quality of all gears had been checked before they were set on the testing device” Have the gears been measured with a CMM device? How did the quality of the gears change before and after the coating?
5. Photographs of gears can be given after tests and interpreted in Section 3. The peeling of the coatings can be proven by photographs.
6. “The sound pressure level of polymer gears without coating is lower than that of polymer gears with different coatings. As these coatings are moderately resistant, they are peeled during operation, and parts of the coating come into the contact point of the meshing area and cause friction and disturbances in the meshing process”. You must prove it with the help of literature or photos of test gears.

Author Response
Comment #1: The pictures in Figure 2 are not clear (especially b and c). Can you add clearer photos?
Response: Figure 2 has been improved as suggested.
Comment #2: Section 2.3 shows the test conditions. In this section, you can also specify the test duration.
Response: Better explanation of the test duration has been added in the revised manuscript (see Section 2.3).
Comment #3: In section 2.3 it is said that the microphone is closer to 100 mm from the test gears. Have you looked at what change happens when it's closer or farther away? If you have, can you briefly explain? How did you determine the 100 mm distance? Did you measure the sound level of the testing device before the gears were running?
Response: By increasing and reducing the distance of the microphone, the sound pressure level changes, and it is decreased with the distance. Therefore, it is required to carry out all the measurements at a certain constant distance. The measurement of sound pressure level depends on several factors, i.e. on the source distance from the microphone, the direction of measurement and the acoustics of the environment. However, the microphone distance can be reduced to a minimum, i.e. it can be very near the field of vibrating surfaces in order to avoid the problems of acoustic background. Getting very near the field of vibrating surfaces also depends on the mechanic source of vibrations and disturbances in the acoustic field between the source and the microphone. In our case, we placed the microphone at a distance of 100 mm, which is within the measuring space/housing protected with sound-proof acoustic foam, which is similar to a semi-anechoic chamber.
Comment #4: In line 186 authors said “The quality of all gears had been checked before they were set on the testing device” Have the gears been measured with a CMM device? How did the quality of the gears change before and after the coating?
Response: Before the measurement, the quality of all gears has been checked with a coordinate measuring machine equipped with additional equipment for measuring gears and with the appropriate software. As we were interested in the sound difference between coated polymer gears and those without coating, we observed coating surfaces. The criteria were stability and the presence of coatings in the tooth flanks. We finished with measurements when the peelling of the coating was more than 80%.
Comment #5: Photographs of gears can be given after tests and interpreted in Section 3. The peeling of the coatings can be proven by photographs?
Response: Photographs of the peeling of coated polymer gears and the appropriate explanation have been added in the revised manuscript.
Comment #6: “The sound pressure level of polymer gears without coating is lower than that of polymer gears with different coatings. As these coatings are moderately resistant, they are peeled during operation, and parts of the coating come into the contact point of the meshing area and cause friction and disturbances in the meshing process”. You must prove it with the help of literature or photos of test gears?
Response: Photos of worn polymer gears with different coatings have been added in the revised manuscript.
Round 2
Reviewer 2 Report
I thank the authors for their precise answers.
I would like to comment -shortly- on the claimed correlation between the peeling and the peculiar spectrogram pattern. The authors show pictures of peeling after 22 minutes (i.e. 1320 seconds) while, on the other hand, the spectrograms cover 2 seconds. Observing 2 seconds of a phenomenon lasting approx 600 times more is not enough to take this correlation as granted. Peeling might be one reason. However, other phenomena/problems (e.g. misalignments, gear machining errors, etc.) can be related to the peculiar spectrogram pattern too.
Author Response
Comment #1: I would like to comment -shortly- on the claimed correlation between the peeling and the peculiar spectrogram pattern. The authors show pictures of peeling after 22 minutes (i.e. 1320 seconds) while, on the other hand, the spectrograms cover 2 seconds. Observing 2 seconds of a phenomenon lasting approx 600 times more is not enough to take this correlation as granted. Peeling might be one reason. However, other phenomena/problems (e.g. misalignments, gear machining errors, etc.) can be related to the peculiar spectrogram pattern too.
Response: The sound signal was obtained in the phase of normal gear operating in the testing device. The sound was measured in the running-in phase and in the wear-in phase of the life cycle during increased surface degradation and wear of coatings of a gear pair. Thus, we have monitored the condition by measuring sound during the entire test. After the running-in phase (approx. 4 minutes), patterns of spectrograms already appeared (to a smaller extent) at the beginning of the wear-in phase of the life cycle. Before the measurement, the quality of all gears was checked with a coordinate measuring machine equipped with additional equipment for measuring gears and with the appropriate software. Thus, the required quality of all the gears and their geometric precision were achieved. In this way, stable geometric condition of gears was obtained in the testing process. The gears were assembled carefully and always in the same way. Thus, the impact of the assembly process as a possible cause of error during the operation of the gear pair was reduced.
On the basis of this, it can be concluded that the impact of all possible geometric deviations of gears is very small as they are within the permitted limitations. Errors, e.g. misalignments, gear machining errors, deformation during assembly, and surface damage, can cause patterns in spectrograms in the form of semi-periodic phenomena, but to a very small extent only. The difference is primarily that there is stochastic energy redistribution within the pulsating area, meaning that we have deformed, split (dual) or serrated forms of pulses.